# Fructan Improves Survival and Function of Cryopreserved Rat Islets

**DOI:** 10.3390/nu13092959

**Published:** 2021-08-25

**Authors:** Takuma Nishino, Takanori Goi, Mitsuhiro Morikawa, Kenji Koneri, Satoshi Terada, Makoto Murakami

**Affiliations:** 1First Department of Surgery, University of Fukui, Fukui 910-1193, Japan; takuman95@yahoo.co.jp (T.N.); mmitsu@u-fukui.ac.jp (M.M.); koneri@g.u-fukui.ac.jp (K.K.); makoto@u-fukui.ac.jp (M.M.); 2Department of Applied Chemistry and Biotechnology, Graduate School of Engineering, University of Fukui, Fukui 910-8507, Japan; terada@u-fukui.ac.jp

**Keywords:** islet cryopreservation, islet transplantation, fructan, serum-free freezing medium, DMSO-free freezing medium

## Abstract

Cryopreservation of pancreatic islets enables their long-term storage and subsequent transplantation; however, post-cryopreservation, islets viability, and functions are reduced to a significant extent. Islet is composed of five cells (α cell, β cell, δ cell, ε cell, and PP cell), and blood vessels that carry the nutrition. Freezing technology of the organization has not developed a good method. This paper is studied using a fructan which has been found to effectively freeze protect a material of the cell. Islet transplantation has been established as an effective means of treating patients with type 1 diabetes. In this study, we demonstrated the effectiveness of using a fructan on the cryopreserved islets by showing valid results for diabetes. Isolated rat islets were cryopreserved using phosphate-buffered saline (PBS) supplemented with different concentrations of fructan and/or dimethyl sulfoxide (DMSO) in FBS. The survival rates of the islets were estimated at different time intervals, and insulin secretion function was tested in vitro. Furthermore, the in vivo function was tested by syngeneic transplantation into streptozotocin-induced diabetic rats, and the grafts were analyzed histologically and immunohistochemically. Fructan significantly increased islet survival; 30% fructan led to survival rates of more than 90% on day 3, which was significantly higher than those of the DMSO groups (*p* < 0.05). For both fructan and DMSO, the survival showed dose dependence, with the highest rates observed for 30% fructan and 10% DMSO, respectively (*p* < 0.05). The fructan groups showed a significantly increased insulin secretion volume in comparison to the DMSO groups (*p* < 0.05). Furthermore, cell clusters of pancreatic islets were well maintained in the fructan group, whereas margin collapse and vacuolation were observed in the DMSO group. Three days after transplantation of pancreatic islets preserved with 30% fructan, the blood glucose levels of diabetic rats were restored to the normal range, and removal of transplanted pancreatic islets from the kidney led to a profound increase in blood glucose levels. Together, these results show that a fructan is effective at cryopreserving rat pancreatic islets for subsequent transplantation.

## 1. Introduction

Pancreas transplantation is an established form of treatment for type 1 diabetes, but due to the significant surgical stress involved, the operation presents a high risk for complications [1]. Pancreatic islet transplantation is a low-invasive tissue transplantation treatment, in which isolated pancreatic islet enzymes are injected into the ducts of a pancreas provided by a donor, followed by isolating only the pancreatic islet tissue through a process of pancreas digestion and pancreatic islet purification, with a final step of intraportal infusion. This last step is required to avoid severe hypoglycemic events in patients with type 1 diabetes and to improve their quality of life. The low invasiveness of this procedure makes it superior to pancreas transplantation, which has led to its wide establishment in Europe and the USA [2]. However, its establishment in Japan has thus far been a challenge owing to a relative lack of donors. Therefore, the creation of islet banks could resolve this situation if the semi-permanent pancreatic islet count and function could be maintained and preserved through cryopreservation. Cryopreservation is commonly used to preserve various cells and tissues, which generally involves dimethyl sulfoxide (DMSO). However, DMSO is cytotoxic, and thus its use should be avoided [3,4,5]. Fructan is a polysaccharide contained in plants, which has been proposed to function as an anti-freezing agent (cold protectant), as well as an osmotic protectant (protection against dryness or acid) or a drought protectant [6]. Fructan is the common name of a polysaccharide consisting of D-fructose. Three types are known: non-insulin type (beta-2,1-linked fructan), levan-type (beta-2,6-linked fructan), and graminan-type (includes both beta-2,1-linked and beta-2,6-linked types). Approximately 15% of natural flowering plants contain fructan, and the majority of fructan is present in regions with changing seasons and in regions with extreme droughts and cold. Fructan is water-soluble, and it is believed that this feature may facilitate adaptation by plants to low temperatures and acid, and to rapid changes in osmotic pressure, by enabling them to easily change the lengths of fructose chains [6]. Ogawa et al. [7] demonstrated the efficacy of Japanese leek (*rakkyo*) fructan as a mammalian cell cryoprotectant. This suggests that *rakkyo* fructan may be applied as a cryopreservation liquid for rat pancreatic islets, and thus contribute to advancements in future long-term preservation techniques for islets. The aim of the present study was to evaluate the efficacy of fructan for the cryopreservation of rat pancreatic islets and to analyze the feasibility of such cryopreserved islets for transplantation, using in vitro and in vivo models. The results were compared with those obtained with DMSO-based cryopreservation methods of pancreatic islets, and the possible synergism between fructan and DMSO was also explored. Finally, histological and immunohistological analysis was performed to evaluate the effects of cryopreservation with fructan and DMSO, and investigate the potential underlying mechanism.

## 2. Materials and Methods

### 2.1. Experimental Animals

Six to eight-week-old male Lewis rats (160–240 g, Charles River Japan, Yokohama, Japan) were used as donors and recipients. Recipients were rendered diabetic through intraperitoneal administration of streptozotocin (60 mg/kg body weight; Sigma-Aldrich, St. Louis, MO, USA). Rats used as transplant recipients showed a non-fasting blood glucose level of >350 mg/dl. Blood glucose levels were measured using a Medisafe automatic analyzer (Terumo, Tokyo, Japan). All study protocols were approved by the animal facility of University of Fukui and were conducted according to animal ethics regulations.

### 2.2. Isolation of Pancreatic Islets

Removal of the pancreas was performed under inhalation anesthesia using isoflurane. Cold Hanks’ balanced salt solution (HBSS; Sigma-Aldrich, St. Louis, MO, USA) containing 1.8 mg/mL of collagenase-L (Nitta Zeratin, Osaka, Japan) and 0.3 mg/mL of dispase II (Godo Shusei, Tokyo, Japan) was injected into the common bile duct to achieve distension of the pancreas. The pancreas was removed and digested for 24 min in a 37 °C water bath. After being washed in culture solution three times, the digested tissue was filtered using 500-μm mesh, as previously described [8]. The digested tissue was mixed with 20 mL of Histopaque (Sigma-Aldrich, St. Louis, MO, USA), and 20 mL of HBSS was slowly poured onto the Histopaque layer. Following centrifugation at 2500 rpm for 20 min, floating islets were collected by pipetting [9]. Isolated islets were washed twice in cold HBSS and placed into culture dishes.

### 2.3. Islet Culture

Islets were cultured in RPMI1640 medium (R8758; Sigma-Aldrich) supplemented with 10 mM of nicotinamide (Wako Pure Chemical Industries, Osaka, Japan) [10], penicillin-streptomycin, and 10% fetal bovine serum (FBS). The islets were cultured at 37 °C in humidified air (5% CO2, 95% air).

### 2.4. Experimental Grouping

Islets were cryopreserved using two methods. Fructan groups were cryopreserved with various concentrations (10%, 20%, 30%, 40%) of fructan (Fukui, Japan) and PBS, whereas the DMSO and FBS groups were cryopreserved with various concentrations (0%, 2.5%, 5%, 7.5%, 10%) of DMSO (Nacalai Tesque, Kyoto, Japan) and FBS as a cryoprotectant.

### 2.5. Cryopreservation

For freezing, day-cultured islets were transferred to cryogenic tubes (BD Falcon, Franklin Lakes, NJ, USA). In the fructan groups, cryopreservation liquid from each group was poured slowly in the tubes on ice at pre-adjusted concentrations. In the DMSO groups, the cryopreservation liquid was placed in the tubes on ice, and the concentration was adjusted by gradually adding DMSO to the tube. The tubes were slowly frozen using a BICELL freezer (Nihon Freezer, Tokyo, Japan) at a rate of −1 °C/min and preserved at −80 °C for three days. For thawing, in the fructan groups, frozen islets were thawed rapidly in a 37 °C water bath and then washed with culture medium three times. In the DMSO groups, the DMSO was removed by incubating for 30 min in RPMI 1640 medium supplemented with 0.75 M of sucrose, which was diluted by adding RPMI medium step-wise on the ice. Washed islets were removed, placed in fresh culture medium, and cultured at 37 °C in humidified air.

### 2.6. Survival of Islets after Thawing

After thawing, 50 islets were cultured and the numbers of surviving islets on day 3 were counted with an inverted microscope. Results obtained from all experiments are shown as mean ± standard deviation and the number of survival islets are shown in the bar chart.

### 2.7. Morphology

Islets cultured after preservation with 30% fructan and 10% DMSO were morphologically examined under an inverted microscope (CKX41, Olympus, Tokyo, Japan) on days 1 and 7.

### 2.8. Insulin Release Assay

Twenty cryopreserved islets on culture day 1 were pre-incubated overnight at 37 °C in 2 mL of RPMI1640 (R1383; sugar free) containing 3.3 mM of glucose. After pre-incubation, islets were sequentially incubated in 2 mL of 3.3 mM glucose medium, 20 mM of glucose medium, and 3.3 mM of medium for 1 h each [10]. The amount of insulin secreted into each culture medium was measured using the rat insulin enzyme-linked immunoassay kit (Morinaga Institute of Biological Science, Yokohama, Japan). The stimulation index was calculated as follows: stimulation index = the amount of insulin following incubation in 20 mM of glucose medium/((amount of insulin after the first incubation in 3.3 mM of glucose medium + the second one)/2).

### 2.9. Islet Transplantation

Approximately 800 islets that were frozen with 30% fructan for 3 days and cultured for 1 day after thawing were transplanted to the left renal capsule of the diabetic rats under anesthesia. The change in blood glucose levels was monitored after the transplant for 28 days, and the transplants were removed during nephrectomy. After nephrectomy, blood glucose levels were continuously monitored for 1 week.

### 2.10. Histology

The graft was paraffin-embedded after fixation with formalin. Sections of 4-μm thickness were stained with hematoxylin and eosin, and examined immunohistochemically with an anti-insulin antibody (AbD Serotec, Raleigh, NC, USA) by the EnVision method (Dako, Kyoto, Japan), according to the manufacturer’s instructions.

### 2.11. Statistical Analysis

The generalized Wilcoxon test was used to evaluate differences in the survival of islets. The Mann–Whitney U-test was used to evaluate differences in insulin secretion and in the stimulation index between the fructan and DMSO groups. Values of *p* < 0.05 were considered statistically significant.

## 3. Results

### 3.1. Survival Rate of Islets

In the fructan groups, the rate of decrease in the numbers of thawed cultured pancreatic islets over time reduced with increasing fructan concentration. Concentrations of 30% fructan or higher resulted in a favorable survival rate, with 95% or more of the pancreatic islets showing a maintained morphology on day 3. (Figure 1). There was no significant difference in islet survival between the 30% and 40% fructan groups (*p* = 0.059), but the 30% fructan group showed significantly higher survival than the 20% fructan group (*p* < 0.001). Furthermore, the 10% DMSO group showed a survival rate of approximately 95% at thawing, and the number of islets decreased on approximately 65% on day 3. Therefore, the 30% fructan group showed significantly higher survival compared to the 10% DMSO group (*p* < 0.001).

### 3.2. Morphology

No significant changes in morphology were observed from day 1 to day 7 after thawing, and cell clusters of pancreatic islets were well-maintained in the 30% fructan group; however, cell cluster margins collapsed and vacuolation was found in the interiors of islets in the DMSO groups (Figure 2).

### 3.3. Insulin Release Assay

In all of the fructan groups, the insulin secretion volume in 3.3 mM and 20 mM of glucose media was significantly higher than that in the DMSO groups. Furthermore, the 30% fructan group showed the highest stimulation index of all fructan groups at 2.65, and the 10% DMSO group showed almost the highest S.I of all DMSO group at 3.13. The difference in the stimulation index were not observed in all of the group (Table 1).

### 3.4. Islet Transplantation

Approximately 800 pancreatic islets frozen with 30% fructan were transplanted to streptozotocin-induced diabetic rats at the first day after thawing (*n* = 6). The blood glucose levels in all diabetic rats fell to within the normal range within one week after transplantation, and was stable thereafter. After removal of the kidneys along with transplanted pancreatic islets, blood glucose levels rose again (Figure 3). No adverse events were found in any case.

### 3.5. Histology

Pancreatic islet renal grafts frozen with 30% fructan were observed under a microscope with HE and insulin staining. Survival was confirmed with no abnormalities within the renal capsule, and insulin staining was positive in all specimens (Figure 4). No differences in morphology were found compared to sections of unfrozen pancreatic islets implanted within renal capsules.

### 3.6. Synergistic Effects of Fructan and DMSO

Based on the results of the fructan-only cryopreservation liquid and FBS + DMSO cryopreservation liquid, we evaluated the possibility of synergistic effects in a cryopreservation liquid containing both DMSO and fructan, using various concentrations of DMSO (2.5%, 5.0%, 7.5%, 10.0%) and 30% fructan solution. Islets in the 30% fructan-only group showed significantly higher survival compared to all of the DMSO-added groups, and no synergistic effects were found (Figure 5). The insulin release assay showed that the stimulation index of the fructan + DMSO groups was greater than that of the fructan-only groups, but the fructan group had a higher insulin secretion volume than the fructan + DMSO groups with both 3.3 mM and 20.0 mM of glucose medium (Table 2).

## 4. Discussion

Pancreatic islet transplantation is performed as a therapeutic method for type 1 diabetes mellitus mainly in Europe and the USA. A five-year follow-up survey of patients receiving pancreatic islet-only transplantation showed that C-peptides were maintained in approximately 80% of cases; however, the non-insulin dependency rate dropped to 71% after one year, 52% after two years, and 23% after three years, with approximately 90% of cases becoming insulin-dependent again after five years [11]. Therefore, multiple pancreatic islet transplantations may be required to achieve a favorable outcome and improve quality of life. However, pancreatic islets are often not readily available owing to a lack of donors, or transplants cannot be performed because of an insufficient number of pancreatic islets from a single donor. Therefore, development of a technique that could limit the fixed amount of pancreatic islets lost from cryopreservation and maintain pancreatic islet function would be useful in eliminating the future shortfalls in pancreatic islets and help to establish an islet bank.

Several methods for the cryopreservation of cultured cells have been developed and tested, with emphasis placed on using cryoprotectants that are highly water-soluble, allowing for a high degree of concentration during freezing, and have no cytotoxicity [12,13]. DMSO is commonly used as a low-molecular-weight cryoprotectant. It easily penetrates cell membranes, and is exemplified by ethylene glycol, trimethylene glycol, methanol, dimethylacetoamide, and glycerol, which each demonstrate cryoprotective action [14]. By contrast, high-molecular-weight cryoprotectants, such as polyethylene glycol, polyvinylpyrolidon, hydroxylethyl starch, dextran, and albumin, show cryoprotective action that stops outside of cells [14,15]. Of these cryoprotectants, glycerol is frequently used as a substitute for DMSO to preserve red blood cells at a concentration of 10%, although DMSO is clearly superior with respect to cell survival rate [16,17]. However, DMSO also shows strong cytotoxicity, and in addition to being used as a cryoprotectant, it is also used as a differentiation inducing agent for cultured cells. Therefore, the use of DMSO is not recommended when aiming for stability during cell storage [18,19,20]. In addition, serum is often used during cryopreservation with DMSO, which poses another problem. When animal serum is used, there is a concern of zootic infection such as bovine spongiform encephalopathy [21]. In addition to its high cost, these disadvantages have motivated the development of a serum-free cryopreservation liquid. Fructan as a natural plant compound shows good potential in this regard. It is assumed that fructan in plants may serve as an anti-freezing agent (cold protectant), as well as an osmotic protectant (protection against dryness or acid) or other protectant against drought. Furthermore, fructan is known for its cell membrane strengthening, stabilization action, anti-allergy effects, and anti-tumor activity [20,22]. In particular, Graminan-type fructan is a polysaccharide stored in plants such as *rakkyo* (Japanese leek) and wheat, and various activities are expected since it has a highly branched structure [23]. The *rakkyo* fructan used in this experiment was graminan-type, which links at a 3:1 ratio of beta-2,1 linking to beta-2,6 linking. It is characterized by the fact that it is easily dissolved in cold water (30% or more solubility in 4 °C water) and is capable of high-pressure steam sterilization; Ogawa et al. [7] demonstrated its efficacy as a mammalian cell cryoprotectant. The key factors contributing to this effect are that it is highly water-soluble and can be highly concentrated during freezing, and that it prevents cell membrane damage from ice crystal formation on the outside of cells by enveloping the membranes like a net [7].

In our study, *rakkyo* fructan was used to determine its potential application as a pancreatic islet preservation liquid using rat pancreatic islets. First, the optimum concentration was determined by comparing survival rates in order to apply fructan to the cryopreservation of pancreatic islets. Fructan concentration-dependent improvement in survival rate was observed, with significantly higher survival at concentrations of 30% or less (*p* < 0.05).

The survival rate immediately after thawing in the 30% fructan group was 97.2 ± 3.6%, and survival decreased gradually but was maintained above 93% after 3 days. Between fructan concentrations of 30% and 40%, the 30% group showed signs of a higher pancreatic islet survival rate, but no significant difference was found. Moreover, 30% is thought to be the optimum concentration for cryopreservation of pancreatic islets with fructan, i.e., the result of ruling out toxicity to pancreatic islets even at high concentrations of fructan.

Next, we compared the effects of 30% fructan (considered to be the optimum concentration) with those of 10% DMSO solution (the commonly used cryopreservation liquid). Although there was no difference in the survival rate between the two groups immediately after thawing, there was a sharp decrease in survival rate for the 10% DMSO group, reaching 64.8 ± 14.6 by the third day. Morphological observations of cultured pancreatic islets after thawing showed that islet cell clusters in the fructan groups were well-maintained even after 3 days, whereas the cell cluster margins had collapsed and vacuolation was found in the islets in the DMSO groups. As mentioned above, fructan is considered to demonstrate a protective effect against the structural collapse of cells due to cold stress by stabilizing the cell membranes, but its action on pancreatic islets (which are cell clusters) is assumed to be due to protection from external stimulation (for islets, not individual cells). On the other hand, although DMSO is commonly used as a cryoprotectant to protect cells from ice crystal formation due to freezing on the inside and outside of cells, it ultimately decreased the survival rate, which is likely a result of penetrating the cell clusters in pancreatic islets leading to their collapse. There was no synergistic effect of the fructan-only group. In addition, addition of the low (2.5%) DMSO concentration to fructan resulted in better survival of pancreatic islets compared to the high (10%) DMSO concentration. This suggests that the cytotoxic effects of DMSO outweigh its benefits as a freezing agent for rat pancreatic islets.

The insulin release assay of pancreatic islet function showed that the stimulation index exceeded 2 in both the 30% fructan group and the DMSO groups, suggesting that the beta cells that escaped apoptosis from the freezing and thawing process were still capable of glucose concentration-dependent insulin secretion. Although there was no difference in the stimulation index between the DMSO and fructan groups, the insulin secretion volume at each glucose concentration was significantly higher in the fructan groups, which might reflect the difference in pancreatic islet survival rates (beta cell survival rate).

After pancreatic islets cryopreserved with 30% fructan were transplanted to diabetic rats, individual glucose levels were normalized in all cases and a stable blood glucose level was achieved. Furthermore, the survival of pancreatic islets transplanted to renal capsules was confirmed microscopically. No notable adverse events were found in rats, even when the low-concentration fructan (0.1%) + 3 mL PBS solution was intravenously injected into the rat tail veins. Although there is potential that fructan might be harmful to the living body, this has not yet been completely ruled out, as data on survival rates suggests that it could be considered safe.

The specific mechanism contributing to fructan’s cryoprotective effect is not clear, but it is clear that fructan as a natural compound which protects cells from cold, drought, acid, and other factors in plants inhabiting environments that are subject to rapid change; these same mechanisms may apply to islets that are aggregations of animal cells. Ogawa et al. [7] used cell models in which a fluorescent material was encapsulated in liposomes to determine the amount of escaping fluorescence in cells cryopreserved with and without 30% fructan. Although the mechanism is still merely assumed at present, the results suggested that *rakkyo* fructan may prevent cell injury from external invasion through its abundance of branched forms [6].

## 5. Conclusions

In conclusion, the results of this study suggest that a cryopreservation liquid containing fructan is extremely effective for the cryopreservation of rat pancreatic islets, compared to conventional cryopreservation liquid containing DMSO + FBS. Since the nutrient fructan was found to work effectively for the extremely difficult preservation of the islets of Langerhans in the pancreas. It was found that nutrient rakkyo fructan, a fructose polymer, works effectively for various mechanisms.

Fructose has been found to contribute to the future development of medical care.

## Figures and Tables

**Figure 1 nutrients-13-02959-f001:**
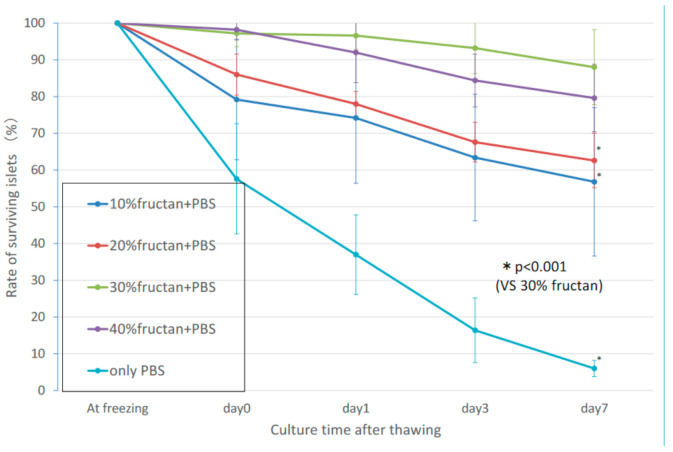
Number of surviving islets in the fructan groups after cryopreservation.

**Figure 2 nutrients-13-02959-f002:**
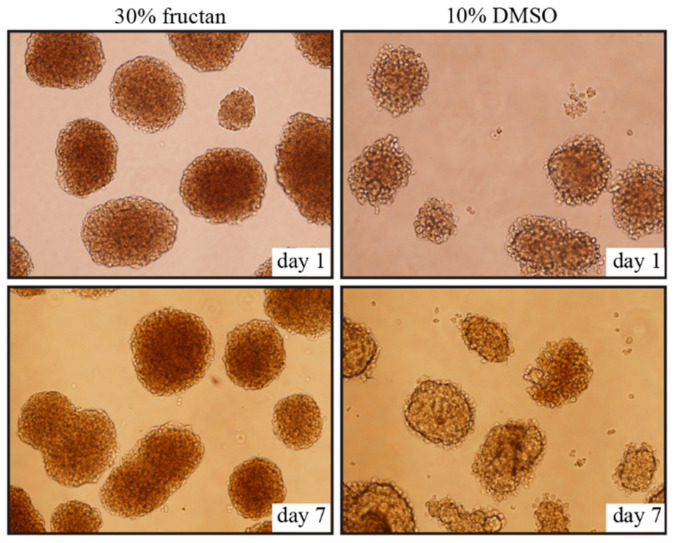
Morphology of the islets cryopreserved in 30% fructan or 10% DMSO after 1 day and 7 days.

**Figure 3 nutrients-13-02959-f003:**
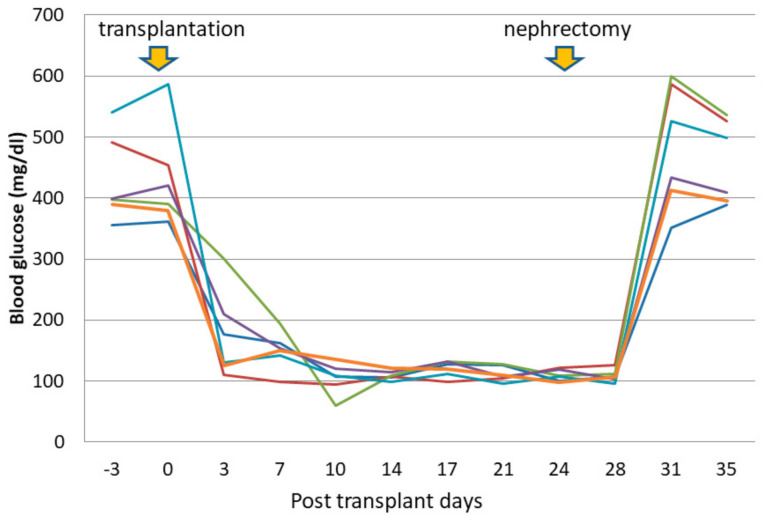
Effects of islet transplantation on blood glucose levels in diabetic rat (*n* = 6).

**Figure 4 nutrients-13-02959-f004:**
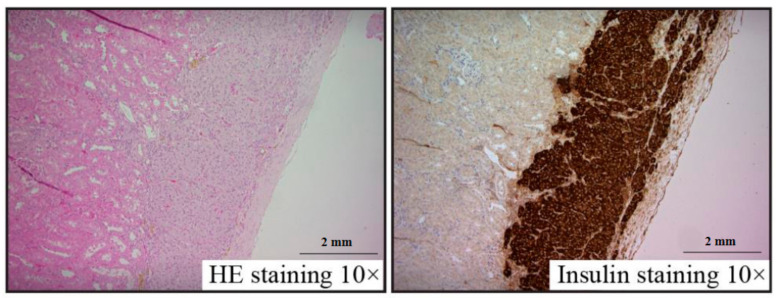
Islet graft histology at 28 days after transplantation.

**Figure 5 nutrients-13-02959-f005:**
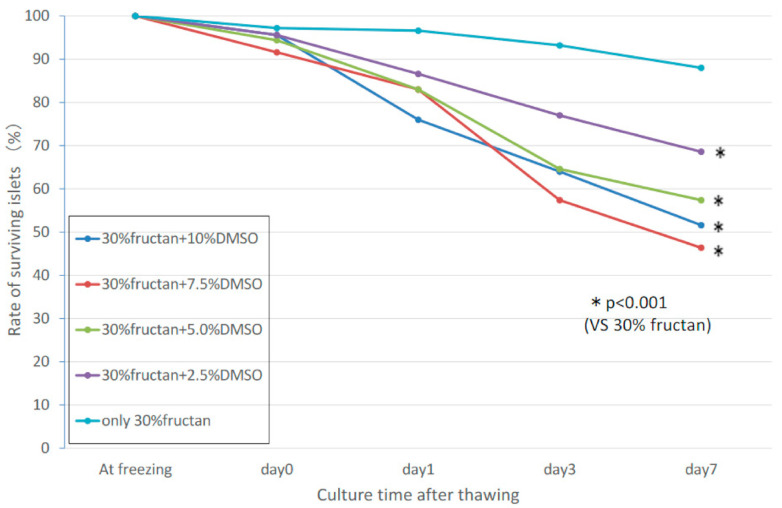
Synergistic effects of fructan and DMSO on islet survival rate during cryopreservation.

**Table 1 nutrients-13-02959-t001:** Insulin release assay.

Insulin Release (*n* = 6)	Low1 (ng/mL/h)	Hi (ng/mL/h)	Low2 (ng/mL/h)	S.I
10%fructan + PBS	7.19 ± 1.13	16.54 ± 2.05	7.13 ± 1.31	2.32 ± 0.35
20%fructan + PBS	7.03 ± 2.81	14.00 ± 3.38	7.61 ± 3.08	2.45 ± 1.00
30%fructan + PBS	6.81 ± 1.41 *	17.97 ± 5.15 *	6.98 ± 1.35 *	2.65 ± 0.62
40%fructan + PBS	5.74 ± 1.90	13.17 ± 1.86	6.07 ± 2.61	2.55 ± 0.95
2.5%DMSO + FBS	5.59 ± 4.90	8.97 ± 4.38	4.31 ± 2.26	2.32 ± 1.48
5.0%DMSO + FBS	3.95 ± 1.09	8.77 ± 2.38	3.91 ± 1.44	2.43 ± 1.14
7.5%DMSO + FBS	4.85 ± 2.18	9.49 ± 2.23	4.84 ± 2.65	2.25 ± 0.72
10%DMSO + FBS	3.66 ± 2.61	9.93 ± 4.26	4.18 ± 2.78	3.13 ± 0.76

* *p* < 0.05.

**Table 2 nutrients-13-02959-t002:** Study of synergistic effects of fructan and DMSO Insulin release assay.

Insulin Release (*n* = 6)	Low1 (ng/mL/h)	Hi (ng/mL/h)	Low2 (ng/mL/h)	S.I
30%fructan + PBS	6.81 ± 1.41 *	17.97 ± 5.15 *	6.98 ± 1.35 *	2.65 ± 0.62
10%DMSO + FBS	3.66 ± 2.61	9.93 ± 4.26	4.18 ± 2.78	3.13 ± 0.76
30%fructan + 10%DMSO	3.81 ± 1.48	11.86 ± 5.85	3.30 ± 2.02	3.11 ± 0.90
30%fructan + 7.5%DMSO	4.91 ± 3.57	12.58 ± 7.58	4.98 ± 3.70	2.92 ± 1.80
30%fructan + 5.0%DMSO	5.09 ± 1.53	11.59 ± 5.04	5.04 ± 1.78	2.25 ± 0.68
30%fructan + 2.5%DMSO	4.31 ± 2.83	12.10 ± 6.30	4.80 ± 3.75	3.31 ± 1.12

* *p* < 0.05.

## Data Availability

The data presented in this study are available on request from the first or corresponding author.

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
