# Peer review of "Fructan Improves Survival and Function of Cryopreserved Rat Islets"

_nutrients, 2021, doi:10.3390/nu13092959_

Round 1
Reviewer 1 Report
Dear Author,
The paper although not containing novel concepts, still has useful information. I recommend doing grammar edits for the abstract and introduction. The methods, results and discussion sections are good. Overall I would recommend its publication.
Here are some of my edits.
Abstract:
Page 1; Line 14-16: Edit the grammar “Islet is the tissue of 5 cells…… Because the….”
Page 1; Line 18: “which have been found to effectiveness” which has been found to effectively
Page 1; Line 20: Edit grammar “we demonstrate the effectiveness of the use of fructan on the cryopreserved islets to 20 prove by showing valid for diabetes.”
Page 1; Line 22: What is DMSO. Define the first time.
Page 1; Line 27: Delete “respectively”
Page 1; Line 35: “fructan is a promising 35 cryopreservation substance for rat pancreatic islets and their subsequent transplantation” to “fructan is effective at cryopreserving rat pancreatic islets for subsequent transplantation.”
Introduction:
Page 1; Line 42: high risk of complications to “high risk for complications”
Page 2; Line 54: dimethyl sulfoxide 54 (DMSO). Full name should be given when it is used the very first time. You have mentioned DMSO several times before this.
Methods:
Page 2; Line 80: 60 mg/kg body weight; how many times injected?
Page 2; Line 87: “of isoflurane” to using isoflurane
Page 3; Line 105: “with two” to using two methods.
Page 3; Line 109: Given that the 30% fructan group and 10% DMSO group showed the best survival 109 rates (see Results), the synergistic analysis was conducted with this combination.
Do not give explanation. In methods, just describe methods used.
Page 3; Line 148: 1day to one day.
Good luck with your research.
Preeti Chhabra
Author Response
Manuscript ID: nutrients-1316504
I greatly appreciate for your suggestions. We made modifications in the main manuscript. We would like to re-submit our manuscript to Nutrients.
Please refer to the text.
We believe that this paper may be considered for the publication in Nutrients.
Takanori Goi M.D., Ph.D
First Department of Surgery, University of Fukui, Japan
23-3 Eiheiji-cho, Yohida-gun, Fukui, 9101193, Japan
Reviewer 2 Report
The study tries to compare the efficacy of fructan vs dimethyl sulfoxide for the cryopreservation of rat pancreatic islets in vitro and in vivo models. The first question I ask myself is whether this article is suitable for Nutrients. I do not find much relationship with NUTRITION. I think it would be more suitable for a magazine in the area of DIABETES. On the other hand, bibliographic citations appear with a numbering different from that of the order of appearance. In the tables you must add below the explanation of the acronyms. The authors only present a reference to the use of fluctan in the discussion. They should compare their results with other studies in the discussion. The discussion lacks the strengths and limitations part of the study.
Author Response
I greatly appreciate for your suggestions.
It was found that nutrient fructan work effectively for various mechanisms. Since the nutrient fructan was found to work effectively for the extremely difficult preservation of the islets of Langerhans in the pancreas, I submit it to "Nutrients".
Nutrient(Fructan) has been found to contribute to the future development of medical care.
And I made modifications in the manuscript. Please refer to the text.
We believe that this paper may be considered for the publication in "Nutrients"
I greatly appreciate for your suggestions again.
Takanori Goi M.D., Ph.D
First Department of Surgery, University of Fukui, Japan.
23-3 Eiheiji-cho, Yohida-gun, Fukui, 9101193, Japan.
Round 2
Reviewer 2 Report
I believe that the article is ready for publication. Although I still think that it does not have much to do with nutrition.